# Unification of a Global Height System at the Centimeter-Level Using Precise Clock Frequency Signal Links

**Ziyu Shen** [1] , **Wenbin Shen** [2,3,*] , **Shuangxi Zhang** [2] , **C. K. Shum** [3] , **Tengxu Zhang** [1] , **Lin He** [1] , **Zhan Cai** [1] , **Si Xiong** [1] **and Lingxuan Wang** [1]

1    School of Resource, Environmental Science and Engineering, Hubei University of Science and Technology, Xianning 437100, China; zyshen@hbust.edu.cn (Z.S.)
2    Time and Frequency Geodesy Center, Department of Geophysics, School of Geodesy and Geomatics, Wuhan University, Wuhan 430079, China
3    Division of Geodetic Science, School of Earth Sciences, Ohio State University, Columbus, OH 43210, USA
*    Correspondence: wbshen@sgg.whu.edu.cn

**Abstract:** The International Association of Geodesy (IAG) aims to establish the International Height Reference System (IHRS) as one of its primary scientific objectives. Central to the realization of the IHRS is global vertical datum unification, which requires the connection of existing local vertical height reference systems (VHS) robustly and consistently. However, conventional methods are not suitable for estimating the offsets between two distant local height systems. In this paper, we propose a framework for connecting two local VHSs using ultraprecise clock frequency signal links between satellites and ground stations, referred to as the satellite frequency signal transmission (SFST) approach. The SFST approach allows for the direct determination of the geopotential and height differences between two ground datum stations without any location restrictions between the two VHSs. The simulation results show that the VHSs of China and the US can be unified with an accuracy of several centimeters, provided that the stability of atomic clocks used on-board the satellite and at on-ground datum locations reaches $4.8 \times 10^{-17} \tau^{-1/2}$ for an averaging time $\tau$ (in seconds). We conclude that the SFST approach shows promise for achieving centimeter-level accuracy in unifying the global vertical height datum and represents a new paradigm for the realization of the IHRS.

**Keywords:** relativistic geodesy; satellite frequency signal transmission; vertical height reference system; global vertical height datum unification

## 1. Introduction

Worldwide, reference frames with long-term stability and homogeneous consistency play crucial roles in accurate modeling or measuring, e.g., gravity fields, the Earth's rotation, geodynamics, and extensive applications, such as global navigation satellite systems, precise positioning, precision orbit determination for satellites or objects in space, time transfer, etc. The International Terrestrial Reference System (ITRS) and its realization, the International Terrestrial Reference Frame (ITRF) [1], provide a globally unified geometric reference frame with accuracy at the millimeter level. However, a commensurate, highly accurate global physical reference frame that uses the Earth's gravity field currently remains elusive [2]. To establish a consistent and accurate physical reference frame, the International Association of Geodesy (IAG) released IAG Resolution No.1. for the definition and realization of an International Height Reference System (IHRS) in 2015 [3], which is a physical world height system for monitoring gravity field variation. Similar to the geometric reference system and frame, the realization of IHRS is the establishment of the International Height Reference Frame (IHRF). The IHRS is defined by the equipotential surface of the Earth's gravity field, where the geopotential at the surface is the conventional value $W_0 = 62{,}636{,}853.4 \text{ m}^2/\text{s}^2$ (zero-height level), and the vertical coordinates are geopotential numbers $C_p = W_0 - W_P$ [2,3]. A key concept of realizing the IHRS is unifying

local vertical height systems (VHSs, which are at the regional scale) worldwide that are interconnected globally. Since local VHSs are usually based on the mean sea level (MSL) determined by tide gauges, and the MSL is not an equipotential surface, different local VHSs exhibit inconsistencies with each other of up to 1~2 m [4]. Identifying the offsets between the origins (datums) of two arbitrary height systems is the main challenge for realizing the IHRS, and various approaches have been tested and discussed.

Currently, four approaches have been discussed extensively and practically applied for height system unification. They are explained briefly as follows with the advantages and drawbacks presented for each.

(1) The conventional approach is leveling with gravity reductions. It is mainly used for the realization of local VHS, and the accuracy can reach a submillimeter level between neighboring leveling points [5]. However, leveling is laborious and time-consuming, while errors accumulate over long distances. In addition, the main drawback of leveling is that it cannot connect two continents separated by the ocean, which makes it impractical for the realization of a global VHS [2].

(2) The second method is oceanic leveling [6]. It is suitable for connecting different local height systems separated by oceans. For example, ocean models can provide a mean dynamic topography correction to elevation datums of countries with coastlines, thus realizing unification. However, oceanic leveling is limited to elevation datums near coastlines. The uncertainty of the oceanographic modeling method could be better than one decimeter [7]. Still, it requires years of continuous observation data from tide gauges with an adequate density distribution to reach this level of accuracy [7], and these are unavailable in many places, such as African areas.

(3) The third method is estimating the anomalous potential by solving the geodetic boundary value problem (GBVP) [8]. It can provide a global solution for height unification, and the precision in well-surveyed regions reaches several centimeters [9–11]. However, in sparsely surveyed regions, the precision drops to the decimeter level [5]. Another drawback of the GBVP method is that it requires prior information of potential or height values from various sources (global geopotential model, tide gauge data, gravity observation data, et al.). The errors in this prior information will influence the precision of the GBVP method. The use of different kinds of prior information in various regions makes it difficult to unify the elevation datums in these regions.

(4) The fourth method is adopting global gravity models (GGMs) with high precision. The EGM2008, for example, is complete with the degree and order of spherical harmonics up to 2159 [12], and we can directly compute the potential $C(P)$ of any given point in the ITRF coordinates by introducing it into the spherical harmonic expansion equation. However, at present, the GGM method has the problem of a precision and resolution trade-off. For instance, the GOCE series models (see, e.g., [13]); also see the products released from ESA (www.esa.int (accessed on 10 March 2023)) and the International Centre for Global Earth Models (icgem.gfz-potsdam.de/ICGEM (accessed on 10 March 2023)) can reach an accuracy level of 1 cm (even higher) but with a poor resolution of $1° \times 1°$. On the contrary, although the EGM2008 model has a relatively high resolution of $5' \times 5'$, its average accuracy is only about 10 to 20 cm [12]. Another drawback of the GGM method is that different models usually present discrepancies because of their different standards and conventions.

Currently, it is challenging to establish IHRF with high precision by any of the approaches described above. Another method, the relativistic geodetic method, has gained increasing attention and discussion in terms of addressing this challenge. The relativistic geodetic method is based on the general theory of relativity [14]: precise clocks at positions with different geopotentials run at different rates. Therefore, the geopotential difference between two arbitrary stations can be measured by precise clocks. The corresponding height propagation based on this method is referred to as "chronometric leveling" [15,16]. Since the relativistic geodetic method requires ultrahigh precision clocks (e.g., for a precision of 1 cm, the stability of clocks should reach $1 \times 10^{-18}$), it was not suitable for practical

applications for a long time because of the limited clock precision. However, with the fast development of high-precision clock manufacturing technology in recent years, optical atomic clocks (OACs), which have uncertainty and accuracy level of around $1 \times 10^{-18}$ and even higher have been developed in various laboratories [17–20]. This guarantees the feasibility of actual applications of the relativistic geodetic methods. Consequently, more and more scientists are paying attention to various potential applications of the relativistic geodetic methods [21–24].

The most precise method for comparing clocks in different places is to connect them via optical fiber links (OFLs) [25]. Therefore, an increasing number of discussions and experiments on clocks connected by OFLs have been carried out and discussed [26–29]. Recently, the most precise measurement in OFL chronometric leveling was conducted by [30]. They used transportable optical clocks with an uncertainty of around $5 \times 10^{-17}$ to determine the geopotential difference between two points in a mountain area between France and Italy. Though their experiments show a height discrepancy of around 20 cm between the observed OFL result and that determined by the conventional approach (leveling and gravity measurement), the $1\sigma$ uncertainty is limited to around 17 m [30]. In addition, Wu et al. proposed a method to unify several local height systems by clock networks connected by OFLs [31]. According to their simulation results, the height systems of the West European region can be unified at a precision of better than 1 cm, under the assumption that the clock frequency uncertainty is $1 \times 10^{-18}$. Although relativistic geodetic methods are now practical and can reach high precision, the adoption of OFL limits its development. The cost of optical fibers will increase rapidly as the distance between two clocks increases or the number of stations in a network increases.

Alternatively, we can compare two clocks by microwave frequency links in space. Even if the two clocks are not intervisible, we can abridge them by a satellite and measure their geopotential difference [32]. This method is regarded as the satellite frequency signal transmission (SFST) approach, which is discussed in detail in [33–35]. Given the assumption that the stability of the OACs is $1 \times 10^{-18}$ within an hour, simulation experiments show that the precision of the geopotential difference between two stations on the ground can reach several centimeters in height [34]. Although its precision is slightly lower than that of the OFL approach, the SFST approach is more convenient and economical. It is promising for use in global VHS unification and IHRS realization.

In this paper, we propose an approach for unifying the global VHS by providing examples of connecting two different arbitrary local VHSs using the SFST approach. In Section 2, we briefly describe the concept of the height reference system and the SFST method. In Section 3, we described the setup of the simulation experiments. Data processing is presented in Section 4. In Sections 5 and 6, we provide the results and conclusions for this work and potential improvements and applications for the future.

## 2. Method

### 2.1. International Height Reference System and Vertical Height Reference System

The International Height Reference System (IHRS) is a geopotential reference system that co-rotates with the Earth, as defined by the International Association of Geodesy (IAG) in 2015 [36]. According to the IHRS, the geopotential on the geoid (simply, the geoidal potential) is a constant value of $W_0$ = 62,636,853.4 m$^2$/s$^{-2}$, and the vertical coordinates are defined as [2]

$$C_P = -\Delta W_P = W_0 - W_P, \tag{1}$$

where $C_P$ is denoted as geopotential number, $\Delta W_P$ is the geopotential difference between the potential $W_P$ at the considered point $P$ and the geoidal potential $W_0$.

A vertical height reference system (VHS) is defined by the geographic elevation or depth compared with a reference surface (which is usually the local sea level) [37–39]. It has a close connection with the concept of the IHRS, because its reference surface can be

a geoidal surface with $W_0$, and the value of $C_P$ (given in m$^2$/s$^{-2}$) can be converted to a physical height $H_P$ (given in m) by the following equation [40,41]:

$$H_P = \frac{C_P}{\hat{g}} = \frac{W_0 - W_P}{\hat{g}}, \tag{2}$$

where $H_P$ can be the orthometric height (OH), normal height, or dynamic height, depending on the types of $\hat{g}$ applied.

The OH is a geometric length measured along the plumb-line from the ground point $i$ ($i = P, Q$) to its corresponding point $i'$ on the geoid ($i' = P', Q'$, see Figure 1). For the OH case, the $\hat{g}$ in Equation (2) is expressed as

$$\hat{g} = \bar{g} = \frac{1}{H_P} \int_0^{H_P} g(h) \, \mathrm{d}H_P, \tag{3}$$

where $\bar{g}$ is the "mean value" of gravity $g(h)$ along the plumb-line. The normal height and dynamic height are approximations of the OH [40], and the OH is, for practical purposes, the height above sea level (in fact, the height above the geoid). In this paper, we regard the OH as the vertical coordinates of the VHS.

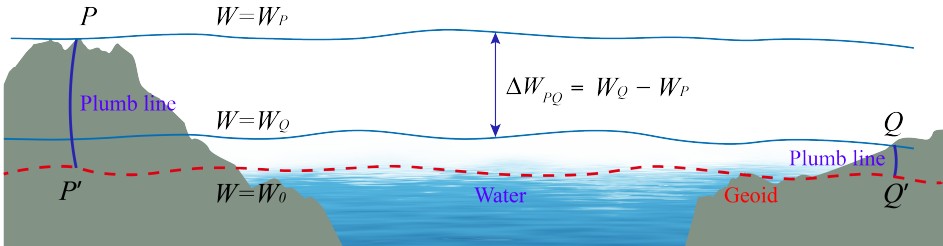

**Figure 1.** The red dashed curve denotes the global geoid, and the two solid blue curves denote the $W = W_P$ and $W = W_Q$ surfaces, respectively. The bold blue curve denotes the plumb-line, along which the height integration is executed.

Currently, it is challenging to unify many different local VHSs, because the global mean sea level is not an equipotential surface. An essential component of realizing the IHRS is to unify the global VHS, which requires a global reference surface that is assumed to be available all over the world to be defined [2].

### 2.2. SFST Method

According to the general theory of relativity, we have the following relationship between the geopotentials $W_P$ and $W_Q$ and the clock frequencies $f_P$ and $f_Q$ at two points $P$ and $Q$ [16,42]

$$\frac{f_P^2}{f_Q^2} = \frac{1 - 2W_Q/c^2}{1 - 2W_P/c^2}, \tag{4}$$

where $c$ is the speed of light in a vacuum. Since the Earth's gravity field is weak, we have the following approximation

$$W_P - W_Q = \frac{f_P - f_Q}{f} c^2 + O(c^{-4}), \tag{5}$$

where $f = (f_P + f_Q)/2$, and $O(c^{-4})$ denote higher order terms that can be neglected if the stations $P$ and $Q$ are stationary near the Earth's surface.

Equation (5) is sufficient for fiber link measurement between two clocks located on the ground. However, the case is much more complex when we use microwave links to compare two clocks located on-board a satellite and at a ground station. For example, a satellite might move at a high velocity, bringing significant Doppler effects. In addition, the ionosphere and troposphere will influence the frequency of microwaves propagating

in space within a medium, and the rotation and tidal effects of the Earth will change the status and environment of a ground station. To address these problems, we formulated the SFST approach for determining the geopotential difference between a satellite and a ground station or between two ground stations [33]. We briefly introduce the main idea and formulations as follows, the details of which refer to [33,34].

Referring to Figure 2, the SFST contains three microwave links. An emitter at a ground station $E$ emits a frequency signal $f_e$ at time $t_1$. When the signal is received by satellite $S$ at time $t_2$, it immediately transmits the received signal $f_e'$ and simultaneously emits a frequency signal $f_s$. The two signals simultaneously transmitted and emitted from the satellite are received by a receiver at ground station $E$ at time $t_3$ and are denoted as $f_e''$ and $f_s'$, respectively. During the emitting and receiving period, the position of the ground station in space changes from $E$ to $E'$. The satellite transmits signals simultaneously as it receives signals, so its position in the signal links is supposed to be point S at time $t_2$. There might be a small amount of latency during transmission, since the satellite's position is slightly different at the time it receives and emits the signals. However, the unsynchronization influence is minimal and can be neglected for the SFST, as explained in [34]). The output frequency shift $\Delta f$ is expressed by a combination of three frequencies as [43,44]

$$\frac{\Delta f}{f_e} = \frac{f_s' - f_s}{f_e} - \frac{(f_e'' - f_e') + (f_e' - f_e)}{2f_e}, \tag{6}$$

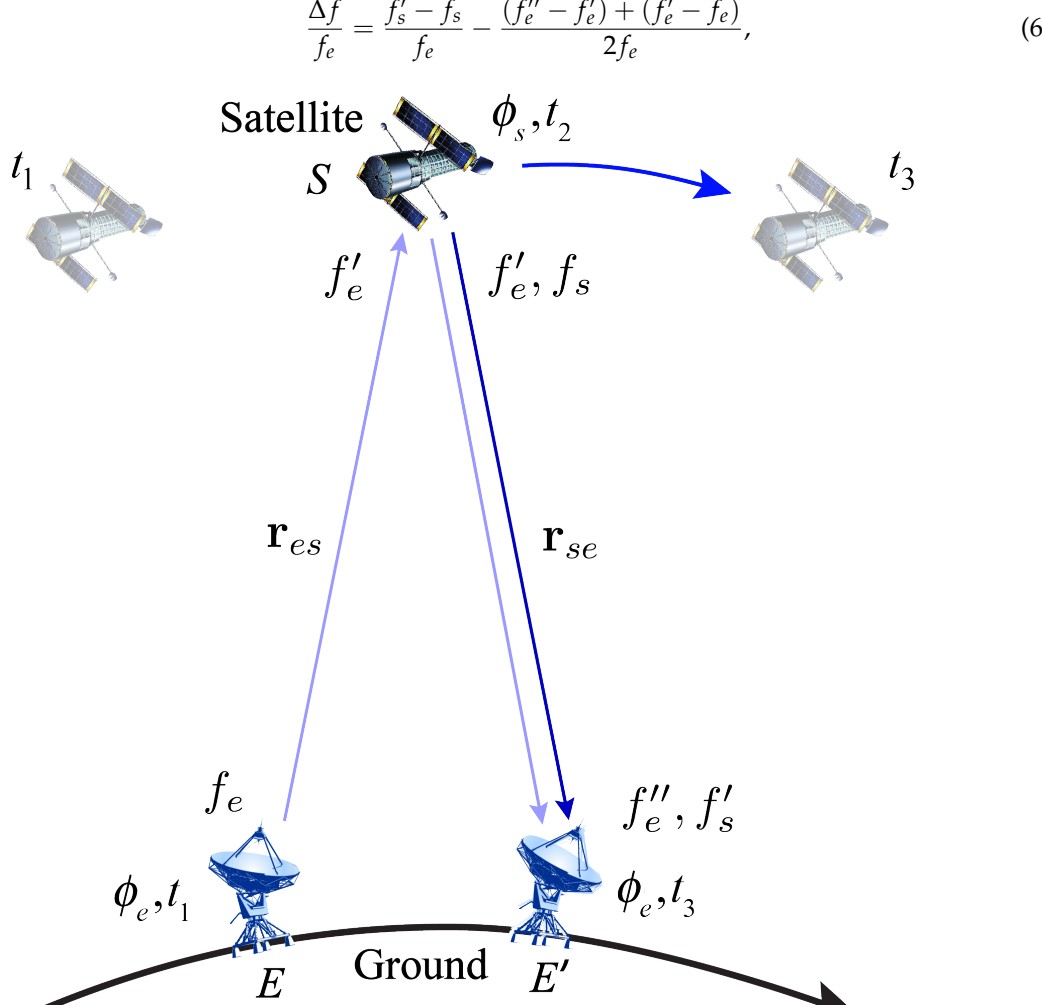

**Figure 2.** Ground station $E$ emits a frequency signal $f_e$ at time $t_1$, denoted by an uplink (blue line). Satellite $S$ transmits the received signal $f_e'$ (the downlink denoted by the blue line) and emits a new frequency signal $f_s$ at time $t_2$ (the downlink denoted by the dark blue line). The ground station receives signals $f_e''$ and $f_s'$ at time $t_3$ at position $E'$. $\phi$ is the gravitational potential (GP), $\vec{r}$ is the position vector.

The beat frequency $\Delta f$, as expressed by Equation (6), cancels out the first-order Doppler effect due to the relative motion between the satellite and the ground station. As for the second-order Doppler effect and the Earth's rotation influence, they are expressed as [43]

$$\frac{\Delta f}{f_e} = \frac{\phi_s - \phi_e}{c^2} - \frac{|\vec{v}_e - \vec{v}_s|^2}{2c^2} - \frac{\vec{r}_{se} \cdot \vec{a}_e}{c^2} + O(c^{-3}), \tag{7}$$

where $\phi_s - \phi_e$ is the gravitational potential difference between the satellite and the ground station; $\vec{v}_e$ and $\vec{v}_s$ are the velocities of the ground station and the satellite (spacecraft), respectively; $\vec{r}_{se}$ is the vector from the satellite to the ground station; $\vec{a}_e$ is the centrifugal acceleration vector of the ground station; and $O(c^{-3})$ denotes terms of higher order than $c^{-2}$. On the right-hand side of Equation (7), the second term denotes the second-order Doppler shift predicted by the special relativity, and the third term represents the effect of the Earth's rotation during the signal's propagation period.

If we omit the higher-order terms $O(c^{-3})$, Equation (7) holds only at the accuracy level of $10^{-15}$ [45]. In this case, we do not need to consider other influential factors, such as the residual ionospheric effects, tidal effect, etc. However, to achieve a one-centimeter level measurement in height, we must consider higher-order terms until $O(c^{-4})$ and various influential factors for satellite–ground microwave links. We derived a theoretical formula that holds at accuracy levels better than $10^{-18}$, which is expressed as [34]

$$\frac{\Delta \phi_{es}}{c^2} \equiv \frac{\phi_s - \phi_e}{c^2} = \frac{\Delta f}{f_e} - \frac{v_s^2 - v_e^2}{2c^2} - \sum_{i=1}^{4} q^{(i)} + \Lambda f + \delta f + O(c^{-5}), \tag{8}$$

where $\Lambda f$ is the sum of all correction terms (it contains corrections of ionospheric and tropospheric effects, tidal effects, and influences of celestial bodies); $\delta f$ is the sum of all error terms; and $q^{(i)}$ ($i = 1, 2, 3, 4$) are quantities related to the positions and velocities of the ground station and satellite, second Newtonian potential, vector potential, and higher-order post-Newtonian terms. The order terms higher than $O(c^{-5})$ are safely omitted. For detailed expressions of the relevant quantities, refer to [34]. Based on Equation (8), when the output frequency shifts, $\Delta f$ is measured, and relevant quantities (such as the position, speed, and acceleration of ground station and satellite) are given, the gravitational potential difference $\phi_{es}$ can be obtained. We also discussed how to determine the geopotential difference between two ground stations that are connected to the same satellite simultaneously [34]. Since the satellite can serve as a "bridge" to connect the two ground sites, the geopotential difference between these two sites can be obtained. Simulation experiments show that the precision of the geopotential difference between two ground sites determined by the SFST method is about 2~5 cm in height [34], under the assumption that the accuracy of the OACs is about $1 \times 10^{-18}$, which is achievable currently [17–19].

### 2.3. Determination of the Height Difference between Two Ground Height Datum Stations

Suppose we have two ground datum stations—the Chinese height datum station at Qingdao and the American height datum station at San Francisco (which were assumed)—denoted respectively as $P$ and $Q$. They are located on two continents that are connected to the same satellite via SFST links simultaneously, cf. Figure 3.

Suppose the gravitational potential difference $\Delta \phi_{PQ}$ between the datum points in China and the US has been determined using the SFST approach described in Section 2.2, then we may determine the geopotential difference $\Delta W_{PQ}$ by the following equation:

$$\Delta W_{PQ} = (\phi_Q - \phi_P) + (Z_Q - Z_P), \tag{9}$$

where $Z_P$ and $Z_Q$ are centrifugal force potentials at $P$ and $Q$, respectively, and $Z$ is expressed as

$$Z = \frac{1}{2}\omega^2(x^2 + y^2), \tag{10}$$

where $\omega$ is the angular velocity of the Earth's rotation, and $x$ and $y$ are coordinates defined in the geocentric Earth-fixed Cartesian coordinate system $o-xyz$ (e.g., ITRF2008, see [1]).

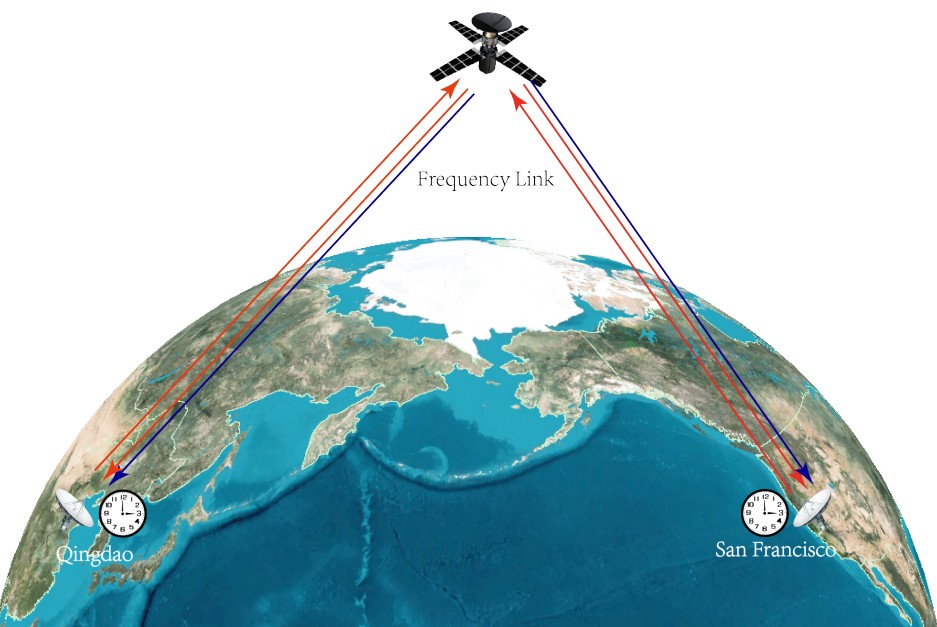

**Figure 3.** Connection of China's HS originating at the Qingdao datum and the US' HS originating at the San Francisco datum via satellite frequency signal transmission.

Suppose the height of point $P$ (denoted as $H_P$) is given, and the geopotential difference $\Delta W_{PQ}$ has been measured by the SFST method; then, the height of point $Q$ (denoted as $H_Q$) can be determined based on Equations (2) and (3), expressed as

$$H_Q = \frac{W_0 - W_Q}{\bar{g}_Q} = \frac{W_0 - W_P - \Delta W_{PQ}}{\bar{g}_Q},$$
$$H_P = \frac{W_0 - W_P}{\bar{g}_P}, \tag{11}$$

where $\bar{g}_P$ and $\bar{g}_Q$ are the average gravity values along the plumb-lines $PP'$ and $QQ'$, respectively (see Figure 1). It should be noted that $\bar{g}_i (i = P, Q)$ cannot be directly calculated by Equation (3), because we do not know the exact density distribution or the gravity distribution $g(h)$ inside the Earth.

We can see that, besides the influence of the given value of $H_P$, the accuracy of the determined $H_Q$ depends on that of $\Delta W_{PQ}$; consequently, it is related to the stability of the optical atomic clocks. Since we cannot determine the "mean value" $\bar{g}(i)$ precisely, in practical applications in plain regions, $\bar{g}_i$ is usually replaced by the following formula [40]

$$\bar{g}_i = g_i + 4.24 \times 10^{-5} H_i, \tag{12}$$

where $g_i$, in gals (cm/s$^2$), is the gravity at ground point $i$, which can be measured by an absolute gravimeter, and $H_i$, in meters, is the height difference between $i$ and $i' (i' = P', Q')$ (see Figure 1). Therefore, according to Equations (11) and (12), we obtain a practical formula for determining $H_Q$, expressed as

$$H_Q = \frac{H_P \cdot (g_P + 4.24 \times 10^{-5} H_P) - \Delta W_{PQ}}{g_Q + 4.24 \times 10^{-5} H_Q}, \tag{13}$$

where $\Delta W_{PQ}$ applies the geopotential unit (m$^2$ s$^{-2}$, 1 m$^2$ s$^{-2}$ = 1000 gal m), and an iteration procedure can be applied if needed.

To connect VHSs, since the heights $H_P$ and $H_Q$ of the two height datum stations are relatively small (say less than 100 m), using Equation (13) is sufficient. For instance, suppose $H_P = 0$, $\Delta W_{PQ} = -100,000$ mGal (which is equivalent to 100 m near Earth's surface), the maximum error caused by using Equation (13) will not exceed 0.4 mm. The reason is stated as follows. In the mentioned case, $|H_Q| = \Delta W_{PQ}/(g_Q + 0.00424)$. The error caused by the uncertainty $\delta \bar{g}_i$ of the chosen mean gravity $\bar{g}_i$ will not exceed 0.00424 gal. Then, we have $|\delta H_Q| = (\Delta W_{PQ}/g_Q^2)\delta \bar{g}_i = 100\,\text{m}\,\delta \bar{g}_i/g_Q \leq 100\,\text{m} \times 0.00424\,\text{gal}/1000\,\text{gal} = 0.4\,\text{mm}$.

Since the heights of site $Q$ and site $P$ are determined under the same basis, these two local VHSs are unified. We can also apply the SFST method to establish a regional height system based on the same principle. The geopotential difference (or height difference) of two arbitrary points in this region can be directly determined, solving the regional height system (regional geoid) tilt problem.

## 3. Experiment Setup

We conducted simulation experiments to verify the SFST method of connecting two VHSs. The main idea of the experiment is to compare a set of true values to a set of simulated observation values, as depicted in Figure 4.

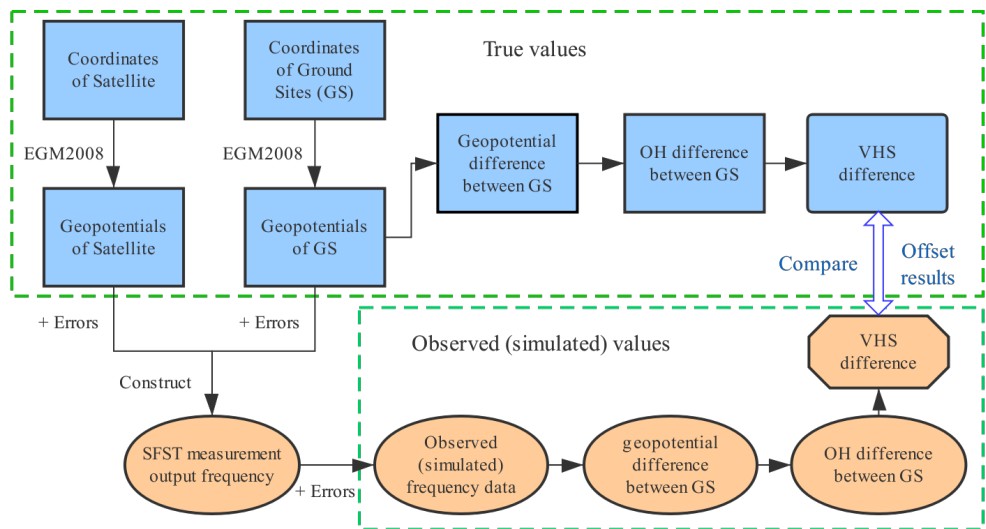

**Figure 4.** The scheme of the simulation experiment.

We chose two datum stations for the experiment. The first was the Qingdao Datum Station (QDDS, located at Qingdao Guanxiangshan mountain, which serves as a height reference datum for China's VHS). The second was the San Francisco Datum Station (SFDS, located at the California Academy of Science, which is supposed to be the height datum station of the US's VHS). The two datum stations are connected via a GNSS-type satellite, as shown in Figure 5. The experiment period was 1.5 h, from 6:00 a.m. to 7:30 a.m. on 1 March 2023. The satellite needed to be intervisible to both ground datum stations during the experiment period; thus, we chose the GPS navigation satellite SVN-56, which satisfies this requirement. It should be noted that the SFST method requires the satellite to be equipped with a frequency signal transponder, which is not applicable in current GPS satellites. However, in our simulated experiments, we assumed that the GPS satellite was equipped with two transponders. Therefore, the satellite could simultaneously connect with the two ground stations with the SFST method. The trace of SVN-56 obtained during our experiment period is depicted in Figure 5.

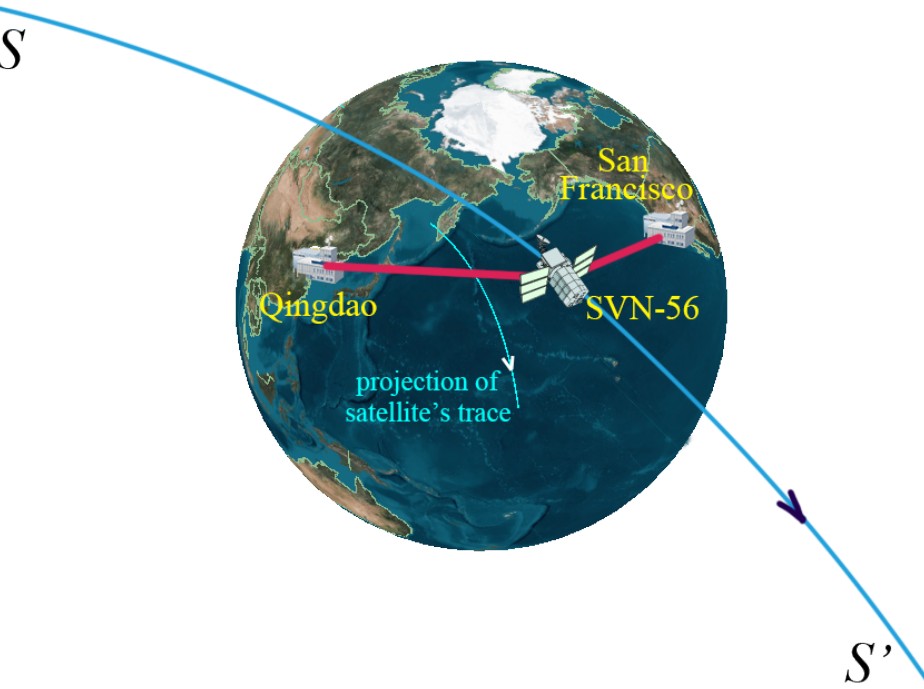

**Figure 5.** Experiments were conducted for the time duration during which the satellite SVN-56 moved from position $S$ to position $S'$ (from 6:00 a.m. to 7:30 a.m., 1 March 2023).

We obtained the orbit information of the GPS navigation satellite SVN-56 from the precise ephemeris provided by IGS (www.igs.org/products (accessed on 20 March 2023)), which are regarded as true values. We sampled the frequency comparison result between the two stations every 5 s. Hence, we obtained a set of observation values for every 5 s. Since the time interval of the precise ephemeris was 15 min, we used polynomial interpolation [46] to acquire the data set in 5 s intervals (true values). Then, we used the EGM2008 model [12] to calculate the gravitational potentials of the satellite and the two ground sites corresponding to the "observation" time points. These gravitational potentials are regarded as true values, because we do not consider the errors caused by EGM2008 (the accuracy of EGM2008 is about 10–20 cm at the ground and better than 1 cm at the GNSS satellite altitude). Then, the true value of the geopotential difference between QDDS and SFDS, $\Delta W_{QD-SF}$, could be obtained.

Microwave signal frequencies are affected by the ionosphere and troposphere medium. We used the International Reference Ionosphere Model [47,48] to obtain the electron density values and the Global Pressure and Temperature (GPT) [49] to obtain the temperature and pressure values. Then, we were able to estimate the ionospheric and tropospheric influences on the signal frequencies [50,51]. The heights and geopotentials of the two ground sites will also be influenced by the periodical tidal effect, which has been well-modeled [52] and can be removed by various software. We used a Python Library Tidal Potential (https://github.com/joernc/tidal-potential (accessed on 10 March 2023)) to calculate the tidal effect.

In our experiment, the two datum stations were connected to SVN-56 simultaneously via SFST links. Relevant input parameters are listed in Table 1. It should be noted that the Chinese government releases the OH of QDDS as China's height datum origin, but the US has no corresponding height datum origin. Therefore, the OH of SFDS was deduced from EGM2008. We assumed the height difference between China's VHS and the US's VHS to be 1.000 m (as true value), and China's VHS was higher than the US's VHS.

**Table 1.** The input data used in the simulation experiments. The coordinates are based on ITRF14

| Entities | | Values of Parameters |
|---|---|---|
| Satellite | ID | SVN-56 (GPS Navigation Sat.) |
| | Coord. | from $(-19{,}167.235509, 3652.729794, 18{,}038.749481)$ |
| | | to $(-26{,}493.102586, 424.868409, 3830.004962)$ |
| Qingdao DS | LLA | $(36.06974°\text{N}, 120.32172°\text{E}, 77.472\ \text{m})$ |
| | ECEF (m) | $(-2{,}605{,}813.108, 4{,}455{,}436.499, 3{,}734{,}494.956)$ |
| | OH (m) | 72.260 |
| San Francisco DS | LLA | $(37.76985°\text{N}, 122.46616°\text{W}, 75.878\ \text{m})$ |
| | ECEF(m) | $(-2{,}709{,}867.959, -4259189.792, 3{,}885{,}328.909)$ |
| | OH (m) | 109.126 |
| Gravity field model | | EGM2008 |
| Ionospheric model | | International Reference Ionosphere |
| Tropospheric model | | Global Pressure and Temperature |
| Tide correction | | Tidal Potential |
| Observation duration | | from 6:00 a.m. to 7:30 a.m., 1 March 2023, |
| Measurement interval | | 5 s |
| Height systems diff. | | 1.000 m (China HS is higher than US HS) |

## 4. Data Processing

According to Equations (8) and (9), the geopotential difference between QDDS and SFDS, $\Delta\hat{W}_{QD-SF}(t)$, can be measured as a time series

$$\Delta\hat{W}_{QD-SF}(t) = \Delta\hat{\phi}_{QD-s}(t) - \Delta\hat{\phi}_{SF-s}(t) + (Z_{SF} - Z_{QD}), \qquad (14)$$

where $\Delta\hat{\phi}_{QD-s}(t)$ and $\Delta\hat{\phi}_{SF-s}(t)$ are, respectively, the observed gravitational potential differences between QDDS and the satellite as well as between SFDS and the satellite at time $t$, and $Z_{QD}$ and $Z_{SF}$ are the centrifugal force potentials of QDDS and SFDS, respectively.

The observed values $\Delta\hat{W}_{QD-SF}(t)$ are different from the true geopotential difference $\Delta W_{QD-SF}$, because various error sources influence them. In this simulation experiment, we considered the clock error $e_{clk}$, ionosphere residual error $e_{ion}$, troposphere residual error $e_{tro}$, and the satellite position and velocity errors $e_{pos}$ and $e_{vel}$, tidal correction residual error $e_{tide}$, and asynchronism error $e_{asy}$. We expect that $\Delta\hat{\phi}_{QD-s}(t)$ and $\Delta\hat{\phi}_{SF-s}(t)$ are measured at the same time $t$, but in practice, they might have slight differences, which will introduce the asynchronism error. The above-mentioned various errors are regarded as noise which is added to the true values. The total errors $e_{all}$ are expressed in the following form:

$$e_{all} = e_{clk} + e_{ion} + e_{tro} + e_{pos} + e_{vel} + e_{tide} + e_{asy}, \qquad (15)$$

The magnitude and behavior of each type of error play important roles in this experiment. Therefore, it is necessary to investigate error models based on these sources to ensure that the simulation accurately reflects the real-world scenario.

Currently, precise OACs have achieved a stability of $4.8 \times 10^{-17}$ at 1 s and $6.6 \times 10^{-19}$ in 1 h for a two clock comparison [19]; therefore, we set the error magnitude of $e_{clk}$ as $4.8 \times 10^{-17}$. Although many kinds of random noises affect OAC signals [53], the most prominent components are white frequency modulation and random walk frequency modulation [54]. Correspondingly, the behaviors of clock errors are modeled as the following equation:

$$e_{clk}(t) = a_{clk} + b_{clk} \cdot t + c_{clk} \cdot \phi(t) + d_{clk} \cdot \int_0^t \xi(t)dt, \qquad (16)$$

where $a_{clk}$, $b_{clk}$, $c_{clk}$, and $d_{clk}$ are constant coefficients, and $\phi(t)$ and $\xi(t)$ are both standard white Gaussian noise. Each term on the right side of Equation (16) has a clear physical meaning; specifically, $a_{clk}$ denotes the initial frequency difference, $b_{clk} \cdot t$ is the drift term, $c_{clk} \cdot \phi(t)$ is the white noise component, and $d_{clk} \cdot \int_0^t \xi(t)dt$ represents the random

walk effect. As we set proper values of constant coefficients following the performance of OACs in [19], we were able to generate a series of frequency comparison data with errors embedded.

For other error sources, their magnitudes are discussed in detail in [34] and listed in Table 2. Most of them are reduced to small magnitudes, since errors in uplink and downlink signals are canceled out in the SFST method. In addition, though the residual influences of the ionosphere and troposphere for the SFST method are at the centimeter level (corresponding to a frequency shift at the $10^{-18}$ level), we established correction models [34] to reduce their influences to the millimeter level. The tidal effects could reach up to 60 cm at their maximum [55]. Still, we can limit the residual error in the vertical direction to 2 mm after corrections for the solid Earth tide [56] and 8 mm for ocean tide loading [57]. The experiment's other time-varying gravity field changes, such as atmospheric loading, ice loading, etc., were not considered, since their residuals are at least one magnitude smaller than the residuals of tidal effects.

Since there are no mature mathematical models for the errors mentioned above, and their influences are much smaller than the clock errors (see Table 2), we adopted a general error model which contains systematic (initial) offset, drift, and white Gaussian noise values for each of the error sources, expressed as the following equation:

$$e_j(t) = a_j + b_j \cdot t + c_j \cdot \phi_i(t), \quad (j = ion, tro, pos, vel, tide, asy) \tag{17}$$

where $a_j$, $b_j$, and $c_j$ are constant coefficients, which are randomly set following the error magnitudes listed in Table 2.

**Table 2.** Error magnitudes of different error sources when determining the gravitational potential difference between a satellite and a ground station. They are transformed to relative frequencies (modified from [34]).

| Influential Factor | (Residual) Error Magnitude in $\Delta f / f_e$ |
|---|---|
| ionospheric correction residual | $\delta f_{ion} \sim 5.5 \times 10^{-19}$ [a] |
| tropospheric correction residual | $\delta f_{tro} \sim 1.9 \times 10^{-19}$ [b] |
| tidal correction residual | $\delta f_{tide} \leq 10^{-18}$ (0.1 m$^2$/s$^2$) |
| position and velocity | $\delta f_{vepo} \sim 3.4 \times 10^{-19}$ (10 mm and 0.1 mm/s [c]) |
| asynchronism | $\delta f_{delay} \sim 10^{-19}$ (below 1 ms) |
| clock error | $\delta f_{osc} \sim 4.8 \times 10^{-17}$ |

[a] 10% of the ionospheric frequency shift after tri-frequency combination; [b] 5% of the tropospheric frequency shift after tri-frequency combination; [c] satellite position errors are assumed to be 10 mm [58], velocity errors are assumed to be 0.1 mm/s [59].

According to Equations (16) and (17), we were able to generate noise signals based on the magnitudes and nature of the error sources at any time. Then, these noises were added to relevant true values, and we obtained a set of relevant "Observed" values, based on which the geopotential difference $\Delta \hat{W}_{QD-SF}(t)$ was determined using Equations (8) and (14). The next step was to convert the geopotential difference to the corresponding height difference. Without a loss of generality, assuming the zero-height surface of China's VHS is just coinciding with the $W_0$ surface, based on China's VHS, the height of SFDS can be calculated by Equation (13), expressed as

$$\hat{H}_{SF}(t) = \frac{H_{QD} \cdot (g_{QD} + 0.0424 H_{QD}) - \Delta \hat{W}_{QD-SF}(t)}{g_{SF} + 0.0424 \hat{H}_{SF}(t)}, \tag{18}$$

where $H_Q = 72.260$ m is the height of the QDDS in China's VHS. In this case, the observed VHS difference between China and US can be obtained as

$$\Delta \hat{H}_{VHS}(t) = \hat{H}_{SF}(t) - H_{SF}, \tag{19}$$

where $H_{SF} = 109.126$ m is the height of the SFDS in the US's VHS, and the unification of the two VHSs can be realized. However, if the zero-height surface of China's VHS does not coincide with the $W_0$ surface (this is, in general, the real case), Equation (18) is not rigorous, and in this case, Equation (11) should be modified to

$$
\begin{aligned}
H_{QD} &= \frac{W_0 + \delta W_{China} - W_{QD}}{\bar{g}_{QD}}, \\
H_{SF} &= \frac{W_0 + \delta W_{US} - W_{SF}}{\bar{g}_{SF}}, \\
\hat{H}_{SF} &= \frac{W_0 + \delta W_{China} - W_{QD} - \Delta W_{QD-SF}}{\bar{g}_{SF}},
\end{aligned}
\tag{20}
$$

where $\delta W_{China}$ is the geopotential difference between the zero-height surface of China's VHS and the $W_0$ surface, and $\delta W_{USA}$ is the geopotential difference between the zero-height surface of the US's VHS and the $W_0$ surface. If $\delta W_{China}$ is unknown, the derived height of the SFDS $\hat{H}_{SF}(t)$ cannot be calculated based on the height of the QDDS $H_{QD}$, even though their geopotential difference is known. Therefore, the height difference between the two VHSs cannot be strictly determined. However, as the $W_0$ surface and a VHS's zero-height surface are close to the mean sea level, and $\delta W_i(i = China, US)$ are relatively small (usually less than $10 \text{ m}^2/\text{s}^{-2}$ [4]); the error introduced by Equations (18) and (19) can be neglected at the current precision level of centimeters. In addition, if we can obtain the value of $\delta W_i$ (which is very promising for the future, see discussions in Section 6), we can also unify the two VHSs based on a rigorous equation. Therefore, for brevity and without a loss of generality, we used Equations (18) and (19) for the height unification calculation. By comparing the observed height difference $\Delta \hat{H}_{VHS}(t)$ and the true difference $\Delta H_{VHS} = 1$ m, we can verify the reliability of the SFST approach for height system unification.

## 5. Results

Since the length of the experiment period (from 6:00 a.m. to 7:30 a.m., 1 March 2023) was 1.5 h, and the measurement interval was 5 s, there were 1080 observation values in total. The results from the first experiment (namely from 6:00 a.m. to 7:30 a.m., 1 March 2023) are shown in Figure 6, with a mean offset of $-3.6$ cm and an STD value of 12.1 cm (see the first row of Table 3).

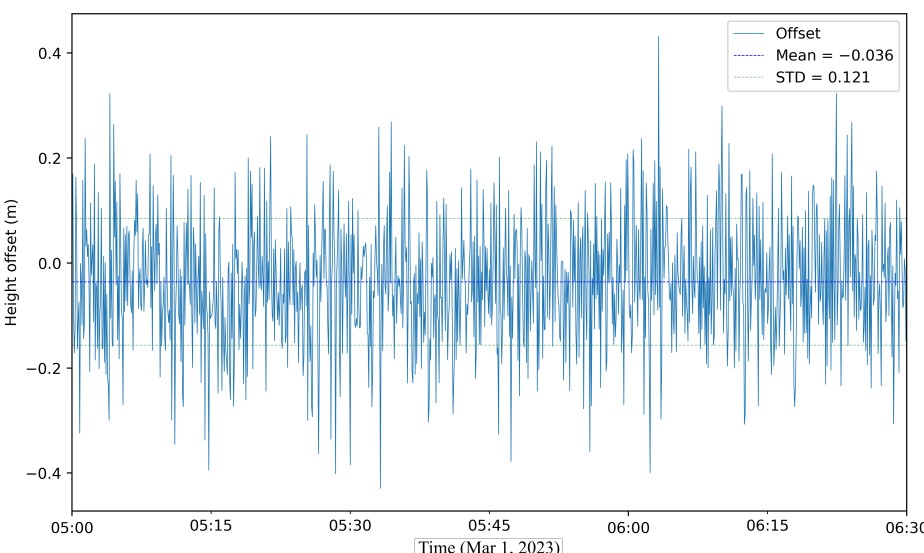

**Figure 6.** The offset between true values and estimated values of the height datum difference determined by the SVN-56 satellite.

We can see that the mean offset was at the centimeter level, while the STD was one magnitude higher at the decimeter level. The reason for this is the main component of the clock errors is white noise, while the drift and random walk effects are not apparent in the results. Since we can significantly improve the stability of the clock after a period (e.g., one hour) of integration, as demonstrated by [19], the height unification accuracy could be improved after a period of integration.

To estimate the reliability of the SFST method, we ran 10 difference simulation experiments with different randomly chosen coefficients, $a_i$, $b_i$, $c_i$ and $d_i$, in Equations (16) and (17) in total. The behaviors of the offset signals were similar. Thus, we only display the mean offsets and STDs in Figure 7. We can see that the mean offsets are limited to the centimeter level, and the most significant mean offset is $-7.5$ cm in the eighth experiment. The final results are listed in Table 3.

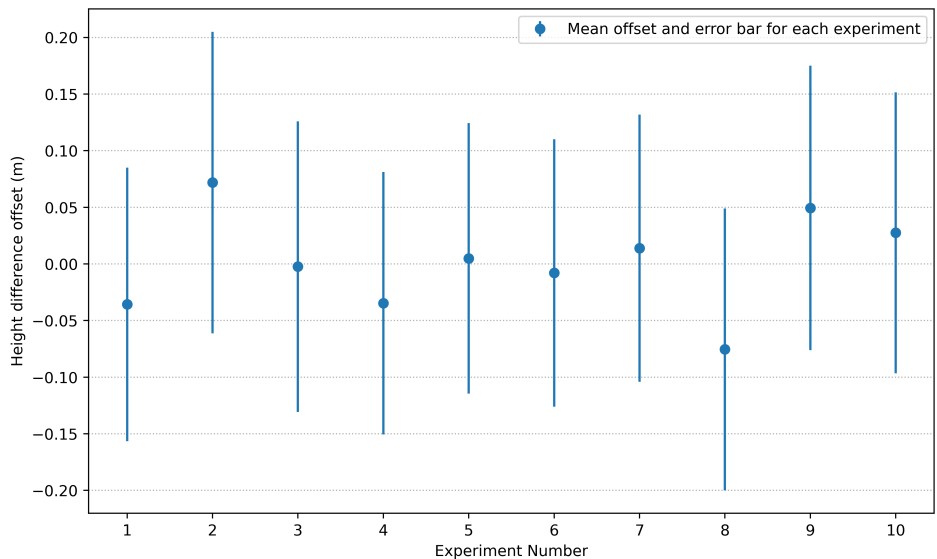

**Figure 7.** The mean offset values and STD values of the height datum difference from 10 different simulation experiments.

**Table 3.** The results of 10 simulation experiments. Relevant parameters are listed in Table 1.

| Experiment No. | Height Diff. between China's VHS and the US' VHS (m) | Offset to True Value (1 m) | STD (m) |
|---|---|---|---|
| 1 | 0.964 | $-0.036$ | 0.121 |
| 2 | 1.072 | 0.072 | 0.133 |
| 3 | 0.998 | $-0.002$ | 0.128 |
| 4 | 0.965 | $-0.035$ | 0.116 |
| 5 | 1.005 | 0.005 | 0.119 |
| 6 | 0.992 | $-0.008$ | 0.118 |
| 7 | 1.014 | 0.014 | 0.118 |
| 8 | 0.925 | $-0.075$ | 0.124 |
| 9 | 1.049 | 0.049 | 0.126 |
| 10 | 1.027 | 0.027 | 0.124 |
| Average | 1.001 | 0.001 | 0.123 |

We may use several days of data to estimate the height difference in practical applications. For instance, we can improve the final results after taking the average of ten different experiments conducted in different periods (e.g., continuous ten days, everyday from 6:00 to 7:30). We may also conduct the experiment via several satellites, such as communication satellites and GNSS satellites, as long as the satellites can simultaneously connect with Qingdao and San Francisco. This could further improve the robustness of the height datum difference result. In the future, we can set up appropriate equipment on multiple satellites

orbiting the Earth and establish continuous SFST links with major benchmark stations on Earth to measure the height differences between stations. This overlapping observation at multiple time periods and from multiple satellites can significantly improve the precision and reliability of the results, thereby achieving centimeter-level high-precision global height system unification.

## 6. Conclusions

This study formulated an approach to unify different local vertical height systems at the centimeter level via ultrahigh precision frequency signal links between one satellite and two datum stations separated by oceans. We performed simulation experiments that connect China's VHS and the US' VHS based on the SFST approach. The results show that the deviation between the true value and the calculated result based on "observations" was around 3 cm with an accuracy level (STD) of about one decimeter in 1.5 h, provided that the OACs' stability achieved a level of $4.8 \times 10^{-17}$ in 1 s. Increasing the experiment period can also improve the accuracy of the SFST approach.

A prerequisite for the SFST approach is frequency synchronization before measurement. The output frequency of the OACs' oscillators should be identical if their locations have the same geopotential value. The error of the initial synchronization will also affect the precision of height unification based on relativistic geodetic methods. When two clocks are separated at a considerable distance, such as in our case, it is very challenging to synchronize them precisely. However, we can realize synchronization by combining a fiber connection and repeated clock transportation. Determining how to precisely synchronize two separated clocks is a particular technical problem that is not entirely relevant to the main topic of this study. Hence, we address that problem in a separate paper.

With the rapid development of time–frequency science, ultrahigh precision OACs (say at $1 \times 10^{-18}$ level or better within one hour) have been developed and are still under improvement, which makes the SFST approach possible for the unification of VHSs at the centimeter-level. The SFST approach is also possible for realizing the IHRS. As a preliminary study, we only connected two stations in this work. However, if a globally covered SFST network is established, the VHSs worldwide could be unified.

The SFST method offers several advantages over conventional methods, primarily its ability to quickly and accurately determine the geopotential difference between any two stations, making it a highly efficient solution. While distance or obstacles, such as mountains or oceans, are not a factor with the SFST method, it does have some limitations. Firstly, it requires ultrahigh precision clocks and appropriate equipment, making measurements relatively tricky. Additionally, its precision is slightly lower compared to optical fiber link methods. Therefore, the best approach is to use the SFST method as a supplement to conventional methods. For instance, we can utilize the SFST method to connect benchmarks of two distant VHSs and use conventional methods or optical fiber links for local VHS unification.

**Author Contributions:** Conceptualization, Z.S. and W.S.; methodology, Z.S.; software, Z.S.; validation, C.K.S. and S.Z.; formal analysis, T.Z.; investigation, L.W.; resources, L.H.; data curation, L.H.; writing—original draft preparation, Z.S.; writing—review and editing, Z.C.; visualization, S.X.; supervision, W.S. and S.Z.; project administration, W.S.; funding acquisition, W.S. and Z.S. All authors have read and agreed to the published version of the manuscript.

**Funding:** This study is supported by the National Natural Science Foundation of China (NSFC) (Grant Nos. 41721003, 42030105, 41631072, 41874023, 42274011, 41974034, 42204006), and the Space Station Project (Grant No. 2020-228).

**Data Availability Statement:** The data obtained in the study are available from the corresponding author upon reasonable request.

**Acknowledgments:** We would like to thank P. Zhang and R. Xu for providing the data patterns of the OACs. We would also like to thank the anonymous reviewers for their valuable comments and suggestions.

**Conflicts of Interest:** The authors declare no conflict of interest.

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
