# Peer review of "Unification of a Global Height System at the Centimeter-Level Using Precise Clock Frequency Signal Links"

_remotesensing, doi:10.3390/rs15123020_

Round 1

Reviewer 1 Report

The authors of the manuscript ”Unification of Global Height System at

Centimeter-level Using Precise Clock Frequency Signal Links”  present a method based on relativistic geodetic approach to establish  the International Height Reference Frame (IHRF), because the gravity geopotential difference between two points on the Earth surface can be measured by precise clocks. 

The height datum unification problem is well introduced by the authors, however I’m not sure to have understood well the experiment setup. At lines 288 and 289 the authors write that “In our experiment, the two datum stations are connected to SVN-56 simultaneously via SFST links”, where SFST means satellite frequency signal transmission (SFST), so I suppose that the SVN-56 satellite has to be connected with the two ground stations (in China and USA) via the SFST links simultaneously, as shown in Figure 2. However, the ground station can not send any signal to the GPS satellite,  except for the  control stations, isn’t it? So in your simulation do you suppose that this connection is possible?

Moreover, which is the most important difference with respect to the paper Shen et al (2017)? The distance between the two ground stations?

Minor remarks:

Line 246:  gal.m —> mGal

Line 288: remove double brackets “))”

Reviewer 2 Report

This is a well-written and interesting paper which can help unify height systems as the technologies in the clocks improve. I recommend for publication after the following minor comments are addressed:

Line 29, change 'physic' to 'physical'.

Line 91, change 'generated' to 'developed'.

Line 187 ~ 190, use semi-colons (instead of commas) to separate the different explanations of the variables in Equation 7. This same comment applies to the other explanations of variables in the equations.

Since there is no Discussion section, I think the Results section is too scanty. Kindly increase the contents in the Results section if possible.

There is no problem with the quality of the English language used in this paper.

Round 2

Reviewer 1 Report

Thanks to the authors for answering my questions. I think the manuscript could be considered for publication.